# The Safety of Negative-Pressure Wound Therapy in Melanoma and Sarcoma Patients: A Systematic Review

**DOI:** 10.3390/jcm14197044

**Published:** 2025-10-05

**Authors:** Silvia Dal Pos, Stela Tafaj, Ilda Hoxhaj, Fortunato Cassalia, Francesco Russano, Saveria Tropea, Paolo Del Fiore, Marcodomenico Mazza

**Affiliations:** 1Soft Tissue, Peritoneum and Melanoma Surgical Oncology Unit, Veneto Institute of Oncology IOV-IRCCS, 35128 Padua, Italy; silvia.dalpos@iov.veneto.it (S.D.P.); marcodomenico.mazza@iov.veneto.it (M.M.); 2Department of Surgical, Oncological and Gastroenterological Sciences (DISCOG), University of Padua, 35128 Padua, Italy; 3Department of Medical and Surgical Sciences, Fondazione Policlinico Universitario Agostino Gemelli, Istituto di Ricovero e Cura a Carattere Scientifico (IRCCS), 00168 Rome, Italy; 4Dermatology Unit, Department of Medicine (DIMED), University of Padua, 35121 Padua, Italy

**Keywords:** negative-pressure wound therapy, melanoma, sarcoma, malignant wounds

## Abstract

**Background/Objectives:** Negative-Pressure Wound Therapy (NPWT) is increasingly used to promote wound healing in chronic and complicated wounds, but its use in surgical oncology is still debated due to theoretical concerns about promoting local tumor recurrence. The aim of this study is to review the available evidence on the oncologic safety of NPWT, specifically regarding the risk of local recurrence in patients undergoing surgery for cutaneous melanoma (CM) or soft tissue sarcoma (STS). **Methods**: A systematic review of the literature was conducted using the MEDLINE/PubMed, EMBASE, and Scopus databases through December 2024 (PROSPERO: CRD42024623405). Case series, retrospective cohort studies, and randomized clinical trials reporting survival data in melanoma and sarcoma patients treated with NPWT were eligible for inclusion. PRISMA guidelines were followed and quality assessment checked. **Results**: Seventeen studies were eligible for the analysis with a total of 285 patients, 197 affected by soft tissue sarcoma and 88 by cutaneous melanoma. The pooled proportion of local recurrence was 5% in patients treated with NPWT, regardless of the histology considered (STS and CM). When comparing NPWT to conventional wound therapy, both the pooled risk ratio (0.87; 95% CI: 0.24–3.11; Tau^2^ = 0.14; I^2^ = 8%) and odds ratio (0.83; 95% CI: 0.20–3.39; Tau^2^ = 0.18) indicated no statistically significant difference in the recurrence rate. **Conclusions**: Current evidence does not suggest an increased risk of local recurrence associated with NPWT in melanoma or sarcoma patients, and mostly, NPWT may have important advantages over standard surgical dressings. More high-power randomized controlled trials with wider follow-up periods are needed to make it possible for practitioners to use this technique without being afraid of higher risk local recurrences.

## 1. Introduction

Since its introduction in managing chronic or complicated non-healing wounds, Negative-Pressure Wound Therapy (NPWT) has been widely validated in the literature for its positive physiological effects on wound healing, in both primary and secondary wound closure. Studies have demonstrated that it reduces edema, enhances blood flow, promotes granulation tissue formation, and facilitates the mechanical contraction of wound edges [1,2]. In addition, considering the improvement in wound healing and the reduction in complications, several studies explore the cost–utility relationship associated with the use of NPWT compared to standard wound dressings. For instance, a randomized trial by Ker et al. reported average cost savings of USD 3903.28 per treatment with an incremental net benefit of USD 1828.27, indicating that NPWT is a cost-effective treatment option for wound care [3]. Nevertheless, the use of NPWT in cancer patients remains contentious due to concerns that it may stimulate neo-angiogenesis, cell migration, and proliferation, factors potentially linked to an increasing risk of local recurrence [4]. Consequently, the current literature includes only a few comparative studies evaluating NPWT versus standard dressings in oncological settings, with even fewer describing survival or local recurrence rates. Notably, a systematic review and meta-analysis by Hays et al. found that NPWT significantly reduced overall surgical site infection and the wound complication rate (*p* = 0.01) following malignant tumor resection, though it did not improve disease-free survival [5]. Similarly, the meta-analysis by Wang et al. reported no statistical evidence of oncological compromise in 118 patients treated with NPWT versus standard wound dressing, when the excision was radical (R0 margins) [6]. Supporting this, a case series by Lik et al. concluded that if the histopathological analysis confirmed complete tumor excision, NPWT did not appear to increase recurrence risk [7]. To further understand the role of NPWT in oncological settings, we conducted a systematic literature review focusing only on its effects on recurrence rates, survival outcomes, overall safety, and feasibility as an adjunct to standard wound care. In particular, special attention was given to patients with cutaneous melanoma (CM) and soft tissue sarcoma (STS), considering the high volume at our referral center and the demolitive implications that this surgery often requires.

## 2. Materials and Methods

### 2.1. Study Strategy

This is a systematic review of studies reporting the application and safety of NPWT in patients with melanoma and soft tissue sarcoma. This review was conducted according to the Preferred Reporting Items for Systematic Reviews and Meta-Analyses (PRISMA) guidelines [8]. The review protocol was registered in PROSPERO (CRD42024623405). Eligible studies were systematically searched on the MEDLINE/PubMed, EMBASE, and SCOPUS databases published up to December 2024. In PubMed, the search strategy used the following terms: *(“Negative-Pressure Wound Therapy”[Mesh]) AND (“Sarcoma”[Mesh] OR “Melanoma”[Mesh])*. The search strategy was tailored to fit the other electronic sources. The retrieved articles from each source were combined, and duplicates were removed. Two investigators (SDP, ST) separately evaluated titles and abstracts, followed by case series, retrospective cohort studies, and randomized clinical trials (RCTs) reporting survival data (overall survival: OS; disease-free survival: DFS) in melanoma and sarcoma patients treated with NPWT, which were eligible for inclusion. Data extracted for each article included the following: study features (year of publication, study design, aim, and sample size), patient and clinical characteristics (sex, age, histology, site, and surgical treatment), and outcome measures (wound and oncological outcomes, local or distant recurrences, OS, DFS, and mean follow-up). Any disagreement was discussed and solved by consensus with a third investigator (MM). Quantitative pooling (risk ratios—RRs; odds ratios—ORs) was restricted to comparative studies. Case reports and case series were included in this review for completeness, but they were analyzed qualitatively only, given their descriptive nature and high risk of bias.

### 2.2. The Assessment of the Quality of the Included Studies

Two investigators (SDP, ST) independently appraised the quality of the included studies according to the critical appraisal tool of the Joanna Briggs Institute (JBI), accordingly [9]. A third investigator (MM) reviewed the assessments, and any discrepancies were resolved by consensus. Twelve studies [10,11,12,13,14,15,16,17,18,19,20,21] were evaluated using the JBI checklist for descriptive studies, while five papers [22,23,24,25,26] were assessed as retrospective comparative studies. Although Shields et al. [26] may be considered an RCT, we evaluated it as a comparative study to simplify the quality assessment. The appraisal focused on several key items, including the clarity of study aims, adequacy of the population or randomization process, appropriateness of statistical analysis, consideration of potential biases, and accuracy of the reported results.

### 2.3. Statistical Analysis

A pooled proportion of local recurrence was calculated using both a fixed-effects model and random-effects analysis, considering the low/moderate statistical heterogeneity between studies. Study weights were assigned based on the number of patients. Heterogeneity among studies was assessed using Cochran’s Q statistic, the I^2^ statistic, and the between-study variance (τ2). I^2^ values of 25%, 50%, and 75% were interpreted as low, moderate, and high heterogeneity, respectively. Sensitivity analysis was performed by excluding studies with extreme outliers or small sample sizes to evaluate the robustness of the pooled estimate. Subgroup analysis was also conducted to compare recurrence rates by tumor type (sarcoma vs. melanoma).

All statistical analyses were performed using RevMan with appropriate meta-analysis packages [27]. Forest plots were generated to visualize individual study proportions and pooled estimates.

## 3. Results

### 3.1. Search Results

A total of 216 non-duplicated articles were identified. After excluding 172 articles based on the title/abstract, 45 potentially eligible full-text articles were reviewed. Of these, 30 were excluded due to different design (n = 10), topic (n = 12), or study population (n = 8). Two more records were identified via hand-search [21,26]. Finally, 17 articles were included in the synthesis [10,11,12,13,14,15,16,17,18,19,20,21,22,23,24,25,26]. The PRISMA flow-chart is shown in Figure 1.

The main aims of the included studies are reported in Appendix A.

### 3.2. Study Characteristics

The patients and tumor characteristics, as well as survival data, of the included studies are summarized in Table 1 and Table 2. Twelve studies [10,11,12,13,14,15,16,17,18,19,20,21] were case reports or case series and were included only in the qualitative synthesis. The remaining five articles [22,23,24,25,26] were comparative studies (standard dressing vs. NPWT) and were eligible for quantitative synthesis. Among these, four were non-randomized trials, while only Shield et al. [26] was an RCT with an adequate methodology and concealed allocation. All studies included a total of 285 patients (min 1–max 62), aged from 34 to 56 years, where 197 patients were affected by an STS, while 88 had CM.

### 3.3. Local Recurrence Rate (LRR)

The pooled proportion of local recurrence was 5% (95% CI: 0–0.16). The overall follow-up periods varied from 6 to 77 months across the included studies.

All reported local recurrences were histologically confirmed. Only in Fourman et al. [20] did recurrence occur in R1 resections (in three cases), while in the remaining studies, the margin status was R0 or not specified.

There was low heterogeneity (I^2^) overall among the included studies. When comparing NPWT to conventional wound therapy, the pooled risk ratio (RR) for local recurrence was 0.79 (95% CI: 0.27–2.31; I^2^ = 8%) using a fixed-effects model, indicating no statistically significant difference. Similarly, using random-effects analysis, the pooled RR for local recurrence was 0.87 (95% CI: 0.24–3.11) with a Tau^2^ = 0.14. The pooled OR was 0.83 (95% CI: 0.20–3.39; Tau^2^ = 0.18), and the risk difference (RD) was −0.03 (95% CI: −0.07–0.02; Tau^2^ = 0). The pooled analyses are shown in Figure 2a,b.

A sensitivity analysis was performed by excluding one study at a time and recalculating the pooled estimate. The results were robust, with no significant differences in the overall effect size. Additional sensitivity analyses excluding studies with small sample sizes or outliers confirmed the stability of the findings. Subgroup analyses showed no significant differences in local recurrence rates between sarcoma and melanoma patients. The pooled proportion of local recurrence in STS and melanoma patients treated with NPWT was 5% and 4%, respectively. The pooled RR was 0.74 (95% CI: 0.06–8.86; Tau^2^ = 1.57; I^2^ = 49%) for melanoma and 1.06 (95% CI: 0.17–6.67; Tau^2^ = 0.28, I^2^ = 15%) for sarcoma.

### 3.4. Quality Assessment

The overall quality of the included studies can be considered moderate, primarily due to the small sample sizes and the prevalent methodology, consisting of case series, case reports, and retrospective studies. A summary brief of quality assessment for descriptive and comparative studies is reported in Table 3 and Table 4. Most of the articles lack a control group and do not provide sufficient evidence to ensure the adequate standardization of treatment protocols. Among the studies with a comparison group, the overall quality remains moderate. While Shields et al. features adequate randomization, the lack of blinding exposes these studies to risks of bias, particularly selection and measurement bias [26]. The positive results reported on the use of NPWT in managing challenging wounds after sarcoma and melanoma excisions highlight the potential of this technique.

## 4. Discussion

This systematic review suggests that NPWT does not appear to increase the risk of local recurrence in patients undergoing resection for CM or STS. The concern that NPWT in cancer patients might stimulate neo-angiogenesis and increase the risk of local recurrence originates from studies on normal tissues. In addition, there is no direct evidence in the literature supporting the notion that NPWT influences oncological progression. As a matter of fact, despite advances in surgical techniques and a multidisciplinary team approach, surgery in STS and CM patients continues to be associated with significant complications, which can have profound impacts on patients’ quality of life [28,29]. Shields et al. showed that the reduction in surgical site infection improved by NPWT after STS excision not only enhances long-term outcomes but also improves patients’ overall experience during cancer treatment [26]. Furthermore, NPWT may offer an important advantage in the healing of radio-treated wounds, due to the notable increase in complication rates [30]. For these reasons, NPWT remains a subject of significant interest, due to the possibility of improving the quality of the postoperative management of these patients. Few studies have evaluated the use of NPWT in STS and CM. Most focused on wound care outcomes, and only seventeen articles, reporting data on survival outcomes, met the final inclusion criteria per the PRISMA guidelines outlined in Figure 2. Of these, six studies examined CM patients, while eleven focused on STS. Currently, two ongoing clinical trials (NCT03175718; NCT02901405) are investigating the effects of NPWT on wound complications after STS resection. However, no trials have explored its impact on CM or assessed its oncological effects. In Appendix A, we report the main aim of the included studies. Except for Bedi et al. [25] and Fourman et al. [20], which consider the evaluation of the local recurrence rate as a primary objective, the other studies reported survival data as secondary outcomes. Our results showed that the pooled proportion of local recurrence in STS and CM patients treated with NPWT is 5% and 4%, respectively. The recurrence rates observed among NPWT-treated patients were comparable to those reported in standard wound management, although the included studies vary widely in terms of design, sample size, and follow-up. While melanoma cases are quite homogeneous in the clinico-histopathological features, for sarcoma patients, several aspects should be considered. One important limitation to have regard to in interpreting these results is the inability to stratify the included patients by histological subtype. The cohort comprises both low-grade and high-grade malignancies, which are known to carry markedly different risks of recurrence. An additional limitation of our review is the incomplete reporting of relevant prognostic variables (such as margin status, further local or systemic treatments, timing to recurrence rate). Importantly, these data were not systematically available in the comparative studies, but also when reported, these factors did not appear to strongly modify recurrence outcomes.

Despite extensive efforts, the local recurrence rate in sarcoma patients remains high, reaching up to 30%, influenced by tumor grade, volume, and histotype, in the case of complete resection [31]. In fact, Dermatofibrosarcoma Protuberans not widely excised has a locally aggressive growth with a high rate of local recurrences (20–50%) [32], while the local recurrence rate could reach 80–90% in other histotypes when simple resection is performed with inadequate or positive margins [33,34,35]. Although no studies have yet demonstrated different effects of NPWT according to histotype, we suggest that tumors of vascular origin may be more susceptible to recurrence by neo-angiogenetic effects induced by negative pressure. In our recent case series, one patient diagnosed with recurrent Epithelioid Angiosarcoma was treated with NPWT following surgery and reported an early local recurrence after five months, with the need to undergo a new wide excision. Since one of the most important benefits of NPWT is its promotion of neo-angiogenesis, this case led us to questioning whether NPWT could induce similar molecular changes in angiosarcoma pathways, increasing the risk of recurrence [36]. Among the five included comparative studies [22,23,24,25,26], 100 patients were treated with NPWT reporting a total of five local recurrences compared to seven reported in the control group (7/158 patients treated with conventional dressing). These findings provide further insight into the oncological safety of NPWT in patients with sarcoma or melanoma, although the limited number of events warrants cautious interpretation. Shields et al. reported data regarding the survival rates considering both local and distant recurrence in 17 STS patients (in 14.3% vs. 30%, respectively, in conventional dressing and NPWT, with a *p* = 0.603). In particular, one patient in the conventional dressing group developed a local recurrence with no metastases; one patient in the NPWT group developed a local recurrence with distant metastases; and two patients in the NPWT group developed metastatic disease with no local recurrence. However, this article does not provide specific data on long-term survival outcomes or recurrence rates associated with either treatment approach [26]. A broader case series with a longer follow-up was, instead, reported by Bedi et al. on 312 STS patients (among which 123 received NPWT). With the numbers available, DFS did not differ between patients treated with or without NPWT (100% [95% CI, 154.09–154.09] versus 96% [95% CI, 152.21–169.16]; *p* = 0.211, respectively) [25]. Regarding CM patients, Seo et al. demonstrated the wound healing efficacy of the combination of NPWT with punch grafting to treat large foot wounds following the excision of acral lentiginous melanoma. The local recurrence rate was 15.4% (2/13) for the patients in the second intention healing group versus 3.6% (1/28) in the NPWT group. The mean follow-up period was 11 months, which cannot be relevant to identify long-term metastasis [24]. However, the strength of the evidence remains limited, and there are several limitations to take into consideration. First of all, the majority of studies were retrospective case series or observational cohort studies, with inherent risks of selection bias and confounding. Only a few studies included control groups, and only one RCT was identified. Furthermore, treatment protocols, follow-up durations, and timing of recurrence were not consistently reported, limiting the comparability of outcomes across studies. This underlying heterogeneity limits the generalizability of pooled estimates, and therefore our results should be interpreted with caution. Another key limitation is the lack of standardization in NPWT application, including timing, pressure settings, and duration of use. These variables may influence both wound healing and potential tumor neo-angiogenesis, yet they are rarely reported in detail. Moreover, the oncologic rationale for caution—based on theoretical risks of local tumor stimulation—has not been substantiated by clinical data so far, but it cannot be definitively excluded. Despite these limitations, this review provides reassuring preliminary evidence that NPWT may be safely applied in carefully selected oncologic patients, particularly in the context of complex or high-risk wounds where traditional closure techniques are not feasible. Future research should aim to clarify the oncologic safety of NPWT through well-designed prospective trials, ideally including standardized protocols, long-term follow-up, and stratification by tumor type and grade. The development of multicenter registries may also help accumulate more robust real-world data.

## 5. Conclusions

The discussion surrounding the use of NPWT in oncological wounds remains a topic of active debate. While current evidence suggests that NPWT is likely safe for cancer patients, definitive conclusions are limited by a lack of high-quality studies. Despite the ongoing controversy, the favorable outcomes reported in managing complex wounds underscore the potential benefits of NPWT in this setting. To establish its oncologic safety and optimize its application in surgical oncology, larger prospective and controlled studies are urgently needed.

## Figures and Tables

**Figure 1 jcm-14-07044-f001:**
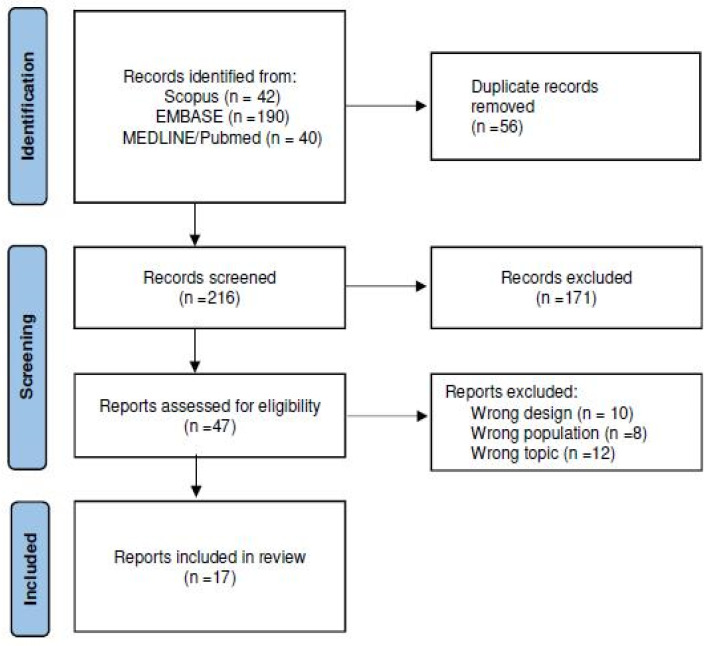
Flow-chart of selection progress using Preferred Reporting Items for Systematic Review and Meta-Analysis (PRISMA) selection process.

**Figure 2 jcm-14-07044-f002:**
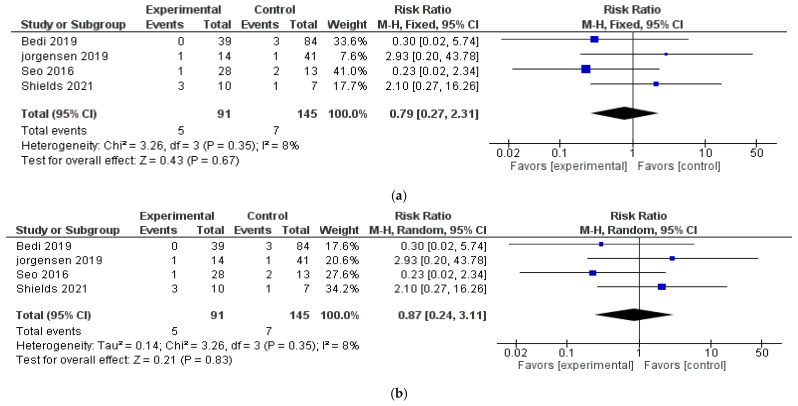
(**a**,**b**). Pooled analyses. (**a**) OR and (**b**) RR of local recurrence rate among included studies [23,24,25,26] that compare negative pressure therapy to conventional dressings.

**Table 1 jcm-14-07044-t001:** Local recurrence rate (LRR) of included case reports or case series [10,11,12,13,14,15,16,17,18,19,20,21]. STS: soft tissue sarcoma; CM: cutaneous melanoma; N/A: not applicable.

Author, Year	Histology	Number of Patients	Median Age (Range)	Margin Status (RO %)	Local Recurrence	LRR (%)	Timing Recurrence (Months)	Mean FU (Months)
Loos B, 2007 [10]	STS	1	56	100	0	0	N/A	18
Senchenkov, A. 2007 [11]	STS	17	65 (42–82)	N/A	1	6	24	N/A
Heller L, 2008 [12]	STS	3	70 (62–76)	N/A	1	33	5	12.3 (9–18)
Agostini T, 2013 [13]	STS	5	47 (20–61)	N/A	0	0	N/A	15.2 (9–24)
Chen Y, 2015 [21]	STS	5	44 (24–68)	100	1	20	12	26 (12–36)
Wu M, 2019 [14]	STS	12	54 (34–69)	100	0	0	N/A	12 (1–24)
Baysal Ö, 2020 [15]	STS	42	39 (8–79)	100	0	0	N/A	30 (5–55)
Lembo F, 2021 [16]	STS	1	34	100	0	0	N/A	6
Miura T, 2021 [17]	CM	5	66 (58–77)	N/A	0	0	N/A	N/A
Korovin, S, 2022 [18]	CM	31	58 (23–86)	N/A	2	6	3	20 (6–35)
Gjorup G.A. 2022 [19]	CM	1	54	100	0	0	N/A	10
Fourman MS. 2022 [20]	STS	62	66 (61–72)	93	5	8.1	33 (15–51)	52 (19–85)

**Table 2 jcm-14-07044-t002:** Local recurrence rate (LRR) of included comparative studies [22,23,24,25,26]. Margin status and timing recurrence refers to NPWT group. STS: soft tissue sarcoma; CM: cutaneous melanoma; NPWT: Negative-Pressure Wound Therapy; N/A: not applicable.

Author	Histology	Margin Status (RO %)	Control Group	NPWT Group	Control Group	NPWT Group	Timing Recurrence (Months)	Mean FU (Months)
			Patients (Median Age)		LRR (%)			
Oh B. H, 2013 [22]	CM	N/A	13 (64.7)	9 (58.3)	0	0	N/A	NA
Jørgensen M. G, 2019 [23]	CM	N/A	41 (57.8)	14 (60)	1	1	N/A	24 (21.9–32.9)
Seo J, 2016 [24]	CM	N/A	13 (64.9)	28 (58.8)	2	1	N/A	77 (62–92)
Bedi M, 2019 [25]	STS	93%	84 (56.5)	39 (54)	3	0	N/A	39.6 (15–98)
Shields D, 2021 [26]	STS	N/A	7 (50)	10 (56)	1	3	N/A	25 (8–42)

**Table 3 jcm-14-07044-t003:** JBI quality assessment for descriptive studies [10,11,12,13,14,15,16,17,18,19,20,21].

ITEMS	Loos [10]	Senchenkov, A. [11]	Heller L [12]	Agostini T [13]	Wu M [14]	Baysal Ö [15]	Lembo F [16]	Miura T [17]	Korovin S [18]	Gjorup G.A. [19]	Fourman MS [20]	Chen Y [21]
Clear criteria of inclusion	+	+	+	+	+	+	+	+	+	+	+	+
Adequacy of the population	+\−	+\−	+\−	+\−	+\−	+	+\−	+\−	+\−	+\−	+\−	+\−
Clear description of characteristics of the participants	+	+	+	+	+	+	+	+	+	+	+	+
Adequacy of data collection	+	+	+	+	+	+	+	+	+\−	+	+	+\−
Relevance of the measurement tools	+\−	+\−	+\−	+\−	+\−	+	+\−	+\−	+\−	+\−	+\−	+\−
Appropriateness of the analysis	+\−	+\−	+\−	+\−	+\−	+	+\−	+\−	+\−	+\−	+\−	−
Biases consideration	+\−	+\−	+\−	+\−	+\−	+\−	+\−	+\−	+\−	+\−	+\−	−
Clarity of the results	+	+	+	+	+	+	+	+	+	+	+	+\−
Conclusion supported by the results	+	+	+	+	+	+	+	+	+	+	+	+\−

**Table 4 jcm-14-07044-t004:** JBI quality assessments for comparative studies [22,23,24,25,26].

ITEMS	Oh B. H [22]	Jørgensen M.G [23]	Seo J [24]	Bedi M [25]	Shields D [26]
Clear criteria of inclusion	+	+	+	+	+
Adequacy of the population	+/−	+/−	+	+	+
Adequacy of comparison of groups (similar treatment/care other than the exposure or intervention or interest)	+/−	+/−	+	+	+
Control group	+	+	+	+	+
Completeness of the measurement tools	+	+	+	+	+
Adequacy of follow up	+	+	+	+	+
Relevance of the outcome measurement tools	+	+	+	+	+
Clarity and reliability of the results and	+	+	+	+	+
Appropriateness of the analysis	+/−	+/−	+	+	+

## Data Availability

Not applicable.

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
