# Peer review of "The Safety of Negative-Pressure Wound Therapy in Melanoma and Sarcoma Patients: A Systematic Review"

_jcm, 2025, doi:10.3390/jcm14197044_

Round 1

Reviewer 1 Report

Comments and Suggestions for Authors

An interesting systematic review conducted in accordance with PRISMA guidelines that evaluates safety, with regards to tumor recurrence, in cutaneous melanoma and soft tissue sarcoma patients treated with negative pressure wound therapy.

The systematic review methodology is clearly explained, and the flow diagram is helpful. 

Both the introduction and discussion are adequate with study limitations described.

The article is clear and well written.

Further review information on:

Manuscript ID: jcm-3860740

The article is a systematic review that evaluates local tumour recurrence in patients with cutaneous melanoma (CM) or soft tissue sarcoma (STS) that have undergone surgical intervention and are then treated with negative pressure wound therapy (NPWT). The article brings together key studies published in the scientific literature and evaluates them to determine if there is an increased risk of local tumour recurrence in CM and STS patients treated with NPWT following surgical removal of the tumours.

The systematic review follows PRISMA guidelines, and the methodology is well explained. The use of the flow diagram is clear and helpful. Quite a wide range of databases have been searched. The key aims of the included articles are listed in supplementary files. Statistical analysis is used to analyse the data from the papers included in the systematic review. I suggest that a statistician reviews the statistical methods performed. Table descriptions could be a bit more detailed e.g., abbreviations could be written in full as well as definitions of the symbols used could be provided.

The study is important because it brings together key information on this topic from the scientific literature, synthesizes and evaluates the information, and performs statistical analysis to arrive at the conclusion that there is not an increased risk of local tumour recurrence associated with NPWT in CM and STS patients that have undergone surgery. This is preliminary but important information. However, the authors also describe limitations of the study and that further research, e.g., high power randomized controlled trials are needed to form definitive conclusions.

In the discussion the authors have sometimes used commas instead of decimal points for numbers. This could be corrected. The conclusions address the main question and the references are appropriate.

Author Response

Reviewer #1 (Comments to the Author):

1.The article is a systematic review that evaluates local tumour recurrence in patients with cutaneous melanoma (CM) or soft tissue sarcoma (STS) that have undergone surgical intervention and are then treated with negative pressure wound therapy (NPWT). The article brings together key studies published in the scientific literature and evaluates them to determine if there is an increased risk of local tumour recurrence in CM and STS patients treated with NPWT following surgical removal of the tumours.

The systematic review follows PRISMA guidelines, and the methodology is well explained. The use of the flow diagram is clear and helpful. Quite a wide range of databases have been searched. The key aims of the included articles are listed in supplementary files. Statistical analysis is used to analyse the data from the papers included in the systematic review. I suggest that a statistician reviews the statistical methods performed. Table descriptions could be a bit more detailed e.g., abbreviations could be written in full as well as definitions of the symbols used could be provided.

The study is important because it brings together key information on this topic from the scientific literature, synthesizes and evaluates the information, and performs statistical analysis to arrive at the conclusion that there is not an increased risk of local tumour recurrence associated with NPWT in CM and STS patients that have undergone surgery. This is preliminary but important information. However, the authors also describe limitations of the study and that further research, e.g., high power randomized controlled trials are needed to form definitive conclusions.

In the discussion the authors have sometimes used commas instead of decimal points for numbers. This could be corrected. The conclusions address the main question and the references are appropriate.

Response:

We thank the reviewer for his valuable suggestions. We have now revised the manuscript, the statistical analysis and the grammar, hence improving the presented article.

Reviewer 2 Report

Comments and Suggestions for Authors

The authors synthesize evidence on the oncologic safety of negative-pressure wound therapy (NPWT) after surgery for cutaneous melanoma and soft-tissue sarcoma, focusing on local recurrence. The clinical question is relevant and timely, and a registered protocol is noted (PROSPERO: CRD42024623405). However, the manuscript has significant issues. With major corrections, the review could contribute meaningfully to the field.

My comments: 

1. The Abstract states the search ran “through December 2024,” whereas Methods §2.1 report searching “through May 2024.” Please reconcile dates across the Abstract, Methods, PRISMA flow, and PROSPERO entry, and state the last search date explicitly. If an update was performed after protocol registration, describe deviations.

2. The manuscript alternately describes the evidence base as “twelve descriptive + five comparative,” “fourteen case reports/series + three RCTs,” and evaluates Shields et al. as a comparative study despite calling it randomized elsewhere. Tables also mix labels.

Please:

- provide a single, consistent study design taxonomy (RCTs, non-randomized comparative cohorts, case series/case reports).

- justify including case reports/series in quantitative pooling (or limit them to qualitative synthesis).

3. You report a pooled local recurrence proportion of 5% (95% CI 0–0.16) and pooled RRs/ORs with “low heterogeneity.” However, the heterogeneity by design and follow-up is substantial (case reports ↔︎ comparative cohorts; follow-up 6–77 months). I² alone may understate clinical heterogeneity; a random-effects model is more appropriate.

4. “Local recurrence” should be operationally defined (site criteria, imaging/pathology confirmation) and time-anchored. Given variable follow-up, pooling proportions may bias results; where available, prefer time-to-event metrics or, at minimum, stratify by follow-up duration and margin status (R0 vs R1). Please standardize definitions and report outcome ascertainment across studies.

5. The risk of local recurrence in STS depends strongly on grade, histotype, size, margins, and radiotherapy; in melanoma, on Breslow depth, stage, site, and closure method. The review should:

- extract and tabulate margin status (R0/R1/R2), adjuvant RT, grade/stage, histotype, and closure strategy when available.

- conduct subgroup/sensitivity analyses (e.g., R0 resections only; with/without adjuvant RT; STS histotypes if reported) to mitigate confounding.

6. You state “minimal heterogeneity” yet also note wide confidence intervals and later report I² = 49% in the melanoma subgroup. Please ensure heterogeneity is reported consistently (overall and subgroup I²/τ²), and justify model choices accordingly.

7. The conclusion that NPWT “does not suggest increased risk” is reasonable as hypothesis-generating, but given small sample sizes, mixed designs, scarce RCT data, and variable follow-up, the certainty is likely low to very low.

8. There are typographical errors and inconsistencies in author names and emails (e.g., “framcesco.russano”; ordering “Del Fiore Paolo”). Please correct per journal style.

9. Ensure reference numbering in Tables 1–2 matches the reference list (e.g., Chen Y (21) vs Oh B.H (21)—appears inconsistent).

Comments on the Quality of English Language

There are several grammatical issues (e.g., “professionists,” “redacted a review”). A thorough English edit will improve readability.

Author Response

Reviewer #2 (Comments to the Author):

  1. 1. The Abstract states the search ran “through December 2024,” whereas Methods §2.1 report searching “through May 2024.” Please reconcile dates across the Abstract, Methods, PRISMA flow, and PROSPERO entry, and state the last search date explicitly. If an update was performed after protocol registration, describe deviations.

Response:

We thank the reviewer for pointing out this typing error,we apologize for the mistake. The correct date for the search is December 2024, therefore we have reconciled dates across abstract, methods and Prospero.

2.The manuscript alternately describes the evidence base as “twelve descriptive + five comparative,” “fourteen case reports/series + three RCTs,” and evaluates Shields et al. as a comparative study despite calling it randomized elsewhere. Tables also mix labels.

Please:

- provide a single, consistent study design taxonomy (RCTs, non-randomized comparative cohorts, case series/case reports).

- justify including case reports/series in quantitative pooling (or limit them to qualitative synthesis).

Response:  We thank the reviewer for this important observation. We revised the classification of the included studies to ensure consistency throughout the manuscript. Studies were now uniformly categorized as follows:

  • Case reports and case series: 12 studies (references 10–21), included only in the descriptive and qualitative synthesis.

  • Comparative studies: 5 studies (references 22–26), eligible for quantitative analysis. Among these, 4 were non-randomized trials, while only Shields et al. (26) was a randomized controlled trial with adequate methodology and concealed allocation.

We also confirm that case reports and case series were not included in the quantitative pooling , but were retained in the review for completeness and qualitative description of outcomes.

The Methods and results sections have been revised accordingly to clarify these points.

  1. You report a pooled local recurrence proportion of 5% (95% CI 0–0.16) and pooled RRs/ORs with “low heterogeneity.” However, the heterogeneity by design and follow-up is substantial (case reports ↔︎ comparative cohorts; follow-up 6–77 months). I² alone may understate clinical heterogeneity; a random-effects model is more appropriate.

Response: We thank the reviewer for this important suggestion. We repeated all comparative analyses using a random-effects model to account for between-study variability. The results remained consistent and confirmed our findings. OR 0.83 (95% CI: 0.20,3.39; Tau² = 0.18); RR 0.87 (95% CI: 0.24, 3.11; Tau² = 0.14); RD -0.03 (95% CI: -0.07, 0.02; Tau² =0).  Although statistical heterogeneity was consistently low, we agree that clinical and methodological heterogeneity (different study designs, small sample sizes, and variable follow-up between 6 and 77 months) remains substantial, and this is now explicitly acknowledged as a limitation in the discussion.

  1. “Local recurrence” should be operationally defined (site criteria, imaging/pathology confirmation) and time-anchored. Given variable follow-up, pooling proportions may bias results; where available, prefer time-to-event metrics or, at minimum, stratify by follow-up duration and margin status (R0 vs R1). Please standardize definitions and report outcome ascertainment across studies.

Response :We thank the reviewer for this important observation. We have now revised the included studies and extracted the data. We confirm that all recurrences reported were histologically confirmed. Only Fourman et al. explicitly linked recurrences to R1 resections (3/5 cases), while in the remaining studies recurrences occurred after R0 resections or not-specified margin status. Timing of recurrence, when available, has been added to the table. We have clarified these aspects in the Results and Discussion.

  1. The risk of local recurrence in STS depends strongly on grade, histotype, size, margins, and radiotherapy; in melanoma, on Breslow depth, stage, site, and closure method. The review should:

- extract and tabulate margin status (R0/R1/R2), adjuvant RT, grade/stage, histotype, and closure strategy when available.

- conduct subgroup/sensitivity analyses (e.g., R0 resections only; with/without adjuvant RT; STS histotypes if reported) to mitigate confounding.

Response: We thank the reviewer for this valuable suggestion. We carefully re-examined all included studies and extracted available data on margin status, further systemic and local treatment, histologic grade/histotype. Unfortunately, these variables were reported inconsistently, and—most importantly—they were not systematically available in the comparative studies (both RCT and non-randomized cohorts), which represent the basis for our quantitative analyses. However, in the few descriptive studies where these factors were reported did not appear to influence recurrence outcomes in a way that would alter our conclusions. We have added a statement in the Discussion to clarify this limitation and to emphasize the need for future studies to systematically report these prognostic variables, in order to reduce confounding and allow for adjusted analyses.

  1. You state “minimal heterogeneity” yet also note wide confidence intervals and later report I² = 49% in the melanoma subgroup. Please ensure heterogeneity is reported consistently (overall and subgroup I²/τ²), and justify model choices accordingly.

Response: Thank you for this observation. The melanoma subgroup showed moderate heterogeneity (I² = 49%). We have now corrected this inconsistency in the results.

  1. The conclusion that NPWT “does not suggest increased risk” is reasonable as hypothesis-generating, but given small sample sizes, mixed designs, scarce RCT data, and variable follow-up, the certainty is likely low to very low.

Response: We have now reviewed the Discussion to clarify this limitation and to emphasize the need for future studies in order to reduce confounding and allow for adjusted analyses. 

  1. There are typographical errors and inconsistencies in author names and emails (e.g., “framcesco.russano”; ordering “Del Fiore Paolo”). Please correct per journal style.

Response: We apologize for this mistake; it has now been corrected.Thank you for bringing this to our attention.

  1. Ensure reference numbering in Tables 1–2 matches the reference list (e.g., Chen Y (21) vs Oh B.H (21)—appears inconsistent)..

            Response: We apologize for this mistake; it has now been corrected.

Comments on the Quality of English Language

There are several grammatical issues (e.g., “professionists,” “redacted a review”). A thorough English edit will improve readability.

            Response: Thank you for your observation. We have now reviewed the quality of English.